# SCORPIO: Serving the Right Requests at the Right Time for Heterogeneous SLOs in LLM Inference

## Abstract

Existing Large Language Model (LLM) serving systems prioritize maximum throughput. They often neglect Service Level Objectives (SLOs) such as Time to First Token (TTFT) and Time Per Output Token (TPOT), which leads to suboptimal SLO attainment. This paper introduces SCORPIO, an SLO-oriented LLM serving system designed to maximize system goodput and SLO attainment for workloads with heterogeneous SLOs. Our core insight is to exploit SLO heterogeneity for adaptive scheduling across admission control, queue management, and batch selection. SCORPIO features a TTFT Guard, which employs least-deadline-first reordering and rejects unattainable requests, and a TPOT Guard, which utilizes a VBS-based admission control and a novel credit-based batching mechanism. Both guards are supported by a predictive module. Evaluations demonstrate that SCORPIO improves system goodput by up to $14.4\times$ and SLO adherence by up to 46.5% compared to state-of-the-art baselines.

## 1 Introduction

Large Language Models (LLMs) are increasingly integral to online applications, powering diverse functionalities such as programming assistance [1], enhanced deep search engines [2], and conversational agents [3]. In order to handle the substantial computational requirements of LLM inference, state-of-the-art serving systems such as vLLM [4] and SGLang [5] employ advanced techniques, including continuous batching [6], paged attention [4], chunked prefilling [7], etc. These optimizations markedly improve inference throughput and resource utilization.

Despite these efficiency improvements, existing LLM serving systems [4, 6, 5, 8] predominantly prioritize maximum throughput, often neglecting the Service Level Objectives (SLOs), such as time-to-first-token (TTFT) and time-per-output-token (TPOT). Typically, these systems greedily admit and serve incoming requests [9], without deep visibility into the SLO requirements of requests. Simultaneously, SLO requirements across different applications are inherently heterogeneous [10]. For instance, programming assistants often demand low latency for real-time response, whereas chatbots might tolerate slightly higher latency as long as the generation rate exceeds human reading speed [11]. However, existing LLM serving systems treat all requests equally in all scheduling stages. This undifferentiated handling leads to suboptimal SLO attainment. These problems necessitate a fine-grained and adaptive scheduling methodology to support heterogeneous SLOs.

In this paper, we focus on designing an SLO-oriented LLM serving system that is tailored for heterogeneous SLOs, with the goal of maximizing both system goodput and SLO attainment. Our key insight is that the inherent heterogeneity of SLOs can be exploited to dynamically schedule the right requests across all scheduling stages (e.g., queue management, admission control, and batch selection), thereby achieving system-level high SLO attainment. From the TTFT perspective, requests

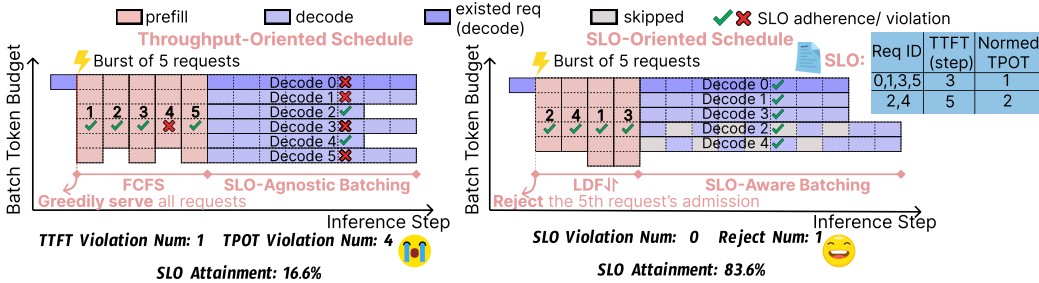

Figure 1: Comparison of throughput-oriented and SLO-oriented scheduling approaches. The throughput-oriented scheduler greedily admits and processes requests without considering per-request SLOs. During the prefill phase, request 4 violates its SLO constraint (4 steps> 3 steps). The SLO-oriented scheduler prevents such violations through least-deadline-first prioritization. During the decode phase, we assume a normalized decode step time as $0.25\times$ BatchSize. The throughput-oriented scheduler batch all requests in each step (BatchSize=6), causing requests 0, 1, 3, and 5 to violate their TPOT constraints (each step consumes a time of 1.5). In contrast, the SLO-oriented scheduler rejects unattainable requests (e.g., request 5) and implements an adaptive fine-grained batching strategy (BatchSize=4). This strategy allows requests with looser TPOT SLOs (requests 2 and 4) to skip certain iterations, ensuring all admitted requests satisfy their TPOT constraints.

with looser SLO can be served a little later. From the TPOT perspective, requests with looser TPOT can skip some iterations of generation, leaving more resources for requests with tighter TPOT.

Based on this insight, we design TPOT Guard and TTFT Guard to handle the heterogeneous TPOT and TTFT SLOs, respectively. For the TPOT Guard, we first define a core concept of TPOT-relative Proportionality (TRP), which quantifies the heterogeneity of TPOT SLOs. Utilizing TRP, we propose a novel VBS-based Admission Control mechanism and a Credit-based Batching mechanism. The former is to control the admission of requests to prevent mass TPOT violations due to overwhelming request ingress. The latter mechanism adaptively batches requests according to their per-request TPOT SLO, achieving system-level high TPOT SLO attainment. For the TTFT Guard, we implement a simple but effective least-deadline-first reordering strategy that prioritizes requests nearing their TTFT deadlines. Additionally, under heavy load, we reject the requests that are unattainable for their TTFT SLOs. To provide decision-making support for these two modules, we develop a predictor module consisting of a sequence length predictor and two analytical models. By orchestrating these complementary mechanisms in concert, SCORPIO enables robust support for diverse workloads.

Our contributions are summarized as follows:

- We identify the critical gap in existing LLM serving systems that prioritize throughput over SLO attainment and propose a scheduling methodology to serve requests with heterogeneous SLOs.
- We propose a TPOT Guard, which consists of a VBS-based Admission Control mechanism and a Credit-based Batching mechanism to provide heterogeneous TPOT SLO guarantees.
- We propose a TTFT Guard, which consists of a least-deadline-first reordering mechanism and an unattainable TTFT SLO reject mechanism to provide heterogeneous TTFT SLO guarantees.
- We develop a predictive module, including a sequence length predictor and two analytical models, which supports the decision-making of the SLO guarantee modules.
- Orchestrating these modules together, we implement SCORPIO. Compared to state-of-the-art baselines, our methods improve the system goodput by up to $14.4\times$ and the SLO adherence rate by up to 46.5% under different scenarios.

## 2 Background and Problem Formulation

A typical LLM generative inference task has two stages: i) the prefill stage, which takes a prompt sequence to generate the first output token; and ii) the decoding stage, which generates new tokens autoregressively. The quality of LLM service is typically evaluated by two key metrics: *time to first token* (TTFT) and *time per output token* (TPOT) [12]. TTFT captures latency for generating

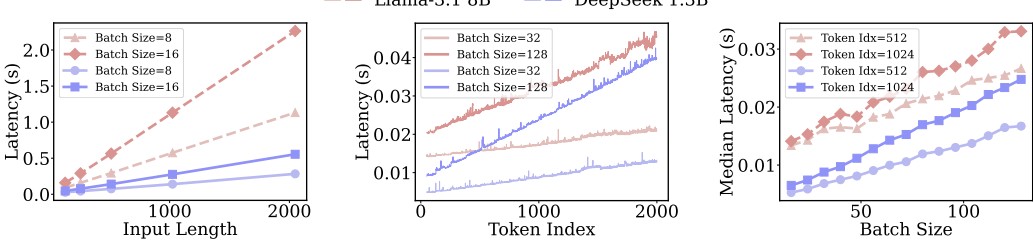

(a) Prefill Latency vs Input Length  (b) Decode Latency vs Token Index  (c) Decode Latency vs Batch Size

Figure 2: Impact of input sequence length (a), batch size (b), and output token index (c) on prefill latency and inter-token latency (ITL).

the first output token after a request is received, while TPOT sets an upper bound on the average latency for generating subsequent tokens. Meeting SLOs plays a key role in providing high-quality LLM service and has been fully researched in other fields like cloud computing [13, 14] and edge computing [15]. However, this problem has not been fully explored in LLM serving. Also, from the perspective of Model as a Service (MaaS) providers, different applications and users have different SLO requirements [10, 16]. This inspires us to explore the heterogeneous SLO attainment problem in LLM serving.

We define the heterogeneous SLO attainment problem as follows: During a time interval $T$, there is a sequence of user requests $\mathcal{R} = \{r_1, r_2, \ldots, r_N\}$ arriving. An LLM inference system processes the requests with a scheduling policy $\pi$. A request $r_i \in \mathcal{R}$ with its TTFT SLO threshold $S_{TT}(r_i)$ and TPOT SLO threshold $S_{TP}(r_i)$ is considered SLO-compliant if it satisfies both $TTFT(r_i) \leq S_{TT}(r_i)$ and $TPOT(r_i) \leq S_{TP}(r_i)$. Let $\mathcal{R}^{good}(\pi) \subseteq \mathcal{R}$ be the subset of those requests that are SLO-compliant. Then the system goodput and SLO adherence rate can be defined as:

$$\text{Goodput}(\pi) = \frac{|\mathcal{R}^{good}(\pi)|}{T} \tag{1}$$

$$\text{Adherence}(\pi) = \frac{|\mathcal{R}^{good}(\pi)|}{|\mathcal{R}|} \tag{2}$$

The objective is to design an online scheduling policy $\pi$ aiming to achieve:

$$\max_{\pi} \quad \left( \lim_{T \to \infty} \mathbb{E}[\text{Goodput}(\pi, T)], \quad \lim_{T \to \infty} \mathbb{E}[\text{Adherence}(\pi, T)] \right) \tag{3}$$

## 3 Method

### 3.1 System Overview

SCORPIO is a system-algorithm co-design framework designed to maximize goodput and SLO adherence. As shown in Figure 3, SCORPIO consists of three key components: 1) predictor, 2) TTFT Guard, and 3) TPOT Guard. When a request arrives, the predictor first predicts the output sequence length, which is used by two analytic models to estimate the TPOT and TTFT. Given the estimated information, the TTFT and TPOT Guards make scheduling decisions accordingly. Specifically, the TTFT Guard first reorders the requests according to their TTFT SLO deadline (i.e., Least Deadline First Reordering (§3.3)). Additionally, it uses the estimated TTFT to reject requests that are unattainable with respect to their TTFT SLOs. Given the new reordered requests, the TPOT Guard uses the estimated TPOT to decide whether to admit the requests into the running batch (i.e., VBS-based Admission Control Mechanism (§3.4)). Then, it employs a Credit-based Batching Mechanism to select which requests to batch in each processing iteration. Lastly, the selected requests are batched and processed in the execution engine.

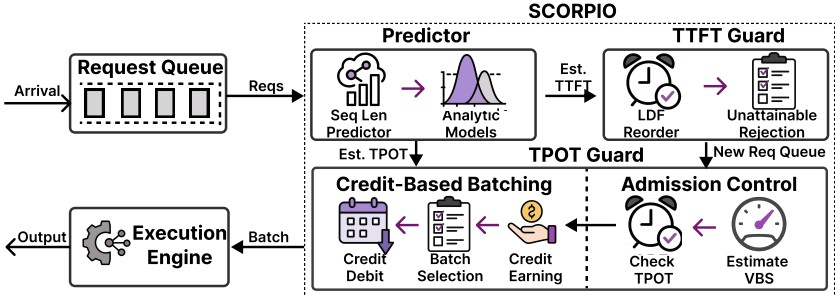

Figure 3: The architecture of SCORPIO.

## 3.2 Predictor

**Sequence Length Predictor.** We employ a lightweight predictor to predict output sequence length. Specifically, we bucketize the potential output length distribution into discrete bins and fine-tune an OPT-125M model [17] as a text classifier to predict the appropriate bin for each sequence.

Unlike prior work [18, 19] that simply set $number\ of\ bins\ =\ 10$ and the bucket size of $\frac{max\ context\ length}{number\ of\ bin}$ to optimize prediction accuracy, we find this coarse-grained binning strategy inadequate for contemporary LLMs. First, it induces a significant class imbalance, resulting in deceptively elevated classification accuracy while demonstrating poor performance in minority length classes. Second, excessively wide bin intervals inherently limit the predictor's resolution, which becomes increasingly problematic as maximum context lengths in modern models continue to grow. Based on a comprehensive analysis of bucketing strategies (§4.4), we find that a medium number of bins (e.g., 100) shows the best tradeoff across several metrics. Model training details can be found in Appendix A.1.

**TPOT Estimator.** To estimate the TPOT of a request, we first observe that the inter-token latency (ITL) is positively correlated with GPU execution state (e.g., batch size and average sequence length) (§2). Therefore, given processing iteration $t$ and the corresponding running requests set $R(t) = \{r_1, r_2, ..., r_n\}$, we develop an analytical model to estimate the ITL:

$$\text{ITL}\{|R|, L_{avg}(R)\} = \alpha \cdot |R| \cdot L_{avg}(R) + \beta \cdot |R| + \gamma \cdot L_{avg}(R) + \delta \tag{4}$$

where $|R|$ represents the running batch size, $L_{avg}(R)$ denotes the average sequence length (including the prompt and generated tokens) of the batch in iteration $t$, and $\alpha, \beta, \gamma, \delta$ are model coefficients determined through empirical measurements. Experiments show that this estimator is highly accurate with an R2 score of over 0.9 (§A.3). Given that a new request $r$ arrives and its predicted output length is $P(r)$, if it is admitted to the running batch $R(t)$, the new set of running requests becomes $R'(t)$, with average length $L_{avg}(R')$. The batch-level TPOT over the next $P(r)$ steps can be derived as:

$$\text{EstimatedTPOT}(|R'|, L_{avg}(R'), P) = \varepsilon \times \{(\alpha \cdot |R'| + \gamma) \cdot (L_{avg}(R') + \frac{P(r)}{2}) + \beta \cdot |R'| + \delta\} \tag{5}$$

Here, to avoid the computational complexity associated with tracking individual request completion times in the batch, we make a simplified conservative assumption: all requests currently in the batch $(r_1, r_2, \ldots, r_n)$ are assumed to continue processing for at least the next $P(r)$ steps. Furthermore, we introduce an inefficiency coefficient $\varepsilon$ ($\varepsilon \geq 1$) to account for potential system overheads [10].

**TTFT Estimator.** Given a waiting requests set $W(t) = \{w_1, w_2, ..., w_n\}$, the TTFT of the request $w_i$ is greater than or equal to the sum of the predicted prefill time of requests in the queue. To estimate the TTFT of a sequence, we first formulate the prefill time of a sequence $w_i$ according to statistical observation (Figure 2a):

$$\text{PrefillTime}(w_i) = \begin{cases} \varphi, & \text{if } L_P(w_i) \leq \theta \\ \alpha * L_P(w_i) + \beta, & \text{otherwise} \end{cases} \tag{6}$$

where $L_P(w_i)$ is the prompt length of request $w_i$, and $\varphi$ is the constant prefill time for sequences with prompt length less than $\theta$. Then, we can estimate the minimal estimated TTFT of a sequence $w_i$ with sorting index $i$ in the waiting queue as:

$$\text{EstimatedTTFT}(w_i) >= \sum_{i=1}^{n} \text{PrefillTime}(w_i) \tag{7}$$

### 3.3 Heterogeneous TTFT Guard

**Least Deadline First (LDF) Reordering.** To handle heterogeneous TTFT SLOs, SCORPIO uses a simple but effective LDF reordering strategy. The deadline for a request $r_i$ is calculated as the time left to its TTFT deadline $S_{TT}(r_i)$. This strategy puts more urgent requests (with earlier TTFT deadline) at the front of the queue, achieving better system-level TTFT attainment.

**Unattainable SLO Rejection.** When the system load is excessively high, some requests will inevitably violate their TTFT SLO and need special handling. In this paper, we tag these requests as unattainable SLO requests and reject them for simplicity. Other strategies, such as processing these requests with a lower priority or migrating them to a different node, are left as future work.

### 3.4 Heterogeneous TPOT Guard

**Key Insight.** As shown in Figure 1, existing methods indiscriminately admit all incoming requests and process them equally in each step. This approach leads to two primary issues: 1) SLO violations for requests with strict TPOTs when contending with those having looser TPOT SLOs, and 2) when the system workload exceeds its processing capacity, greedily serving all requests causes a cascading effect where all requests fail to meet their SLOs. To address the first issue, we design a novel batching mechanism that provides differentiated batching opportunities based on a request's TPOT SLO (i.e., Credit-based Batching Mechanism). Requests with looser TPOT SLOs are offered fewer credits (opportunities) for batching, thereby skipping some execution iterations and leaving more resources for requests with tighter TPOT SLOs. For the latter issue, inspired by load control techniques in cloud computing [20, 21], we introduce a novel admission control mechanism that accounts for the heterogeneity of TPOT SLOs. On one hand, requests estimated to cause TPOT violations are rejected. On the other hand, since a request with looser TPOT SLOs is served intermittently, it can be regarded as a partial request when calculating the batch size (i.e., Virtual Batch Size) for TPOT estimation (§3.2). These two mechanisms together provide TPOT guarantees, as illustrated in Algorithm 1.

**Credit-based Batching** offers requests of different TPOT SLOs with different credits (opportunities) for batching each iteration as mentioned above. To decide how many credits a request earns, we first introduce a concept called TPOT-relative Proportionality (TRP):

**Definition 1** (TPOT-relative Proportionality (TRP)). *Let $S_{TP}(r)$ denote the TPOT SLO of request $r$. Given current processing iteration $t$ and the running requests set $R(t)$, the TPOT-relative Proportionality (TRP) of a request $r \in R(t)$ is defined as:*

$$TRP(r) = \frac{\min_{r \in R(t)} S_{TP}(r')}{S_{TP}(r)}$$

The TRP quantifies the urgency of a request $r$ to be batched compared to admitted requests with the strictest TPOT SLOs. Note that the TRP of a request adaptively responds to changing workloads. Therefore, each request earns credits at its TRP. Accumulating sufficient credit ($\geq 1.0$) grants a request to be batched in the next processing batch. Its credit is then decremented by 1.0, representing the consumption of one processing opportunity. Formally, in each batching step $t$, the following actions are taken:

- **Credit Earning:** For every request $r \in R(t)$, its credit is updated based on its TRP rate:

$$C_r(t) \leftarrow C_r(t) + TRP(r)$$

where $C_r(t)$ is the credit of request $r$ at step $t$, which is initialized as 0.

**Algorithm 1:** TPOT Guarantee Mechanism

---

**Input:** LLM model $M$, Sequence Length Predictor $P$, LDF Sorted Waiting Queue $W$, Running Queue $R$

**Output:** Batched requests set $B$

1 **while** True **do**
2      $B \leftarrow \emptyset$
3      **foreach** $w \in W$ **do**
4          $R' \leftarrow R \cup \{w\}$
5          **if** EstimateTPOT$(VBS(R'), L_{\mathrm{avg}}(R')) \leq \min_{r \in R'} S_{TP}(r)$ **then**
6             $B \leftarrow B \cup \{w\}, W \leftarrow W \setminus \{w\}$ , $R \leftarrow R'$        *// Admission Control*
7          **end**
8      **end**
9      **foreach** $r \in R$ **do**
10          $C_r(t) \leftarrow C_r(t) + TRP(r)$                  *// Credit Earning*
11          **if** $C_r(t) \geq 1.0$ **then**
12             $B \leftarrow B \cup \{r\}, C_r(t) \leftarrow C_r(t) - 1.0$     *// Batch Selection and Credit Debit*
13          **end**
14      **end**
15 **end**

---

- **Batch Selection:** The batch $\mathcal{B}(t)$ is formed by including all requests whose credit, after accumulation, is greater than or equal to the threshold:

$$\mathcal{B}(t) = \{r \in R(t) \mid C_r(t) \geq 1.0\}$$

- **Credit Debit:** For every request $r$ included in $\mathcal{B}(t)$, its credit is decremented by 1.0:

$$\text{If } r \in \mathcal{B}(t), \text{ then } C_r(t) \leftarrow C_r(t) - 1.0$$

This mechanism ensures that over many steps, the frequency of a request $r$ being batched will converge towards its TRP rate.

**VBS-based Admission Control.** When a new request $r$ arrives, to guarantee TPOT SLOs, an intuitive approach is to admit requests into the running batch if admitting it will neither violate its own TPOT SLO nor cause other running requests to violate their TPOT SLOs. However, since credit-based batching causes some requests to skip some execution iterations as mentioned above, directly using the number of running requests as batch size overestimates the actual system load. Since the request $r$ can earn a $TRP(r)$ opportunity to be batched in each iteration if admitted, it can be regarded as a virtual $TRP(r)$ request. Let $R(t)' = R(t) \cup \{r\}$, we can project the actual load of the system as the sum of the TRP of all requests, which we denote as virtual batch size (VBS):

$$\text{VBS}(R') = \sum_{r \in R'} TRP(r) \tag{8}$$

The request $r$ is admitted to the Running Queue at step $t$ if adding it would not cause the estimated TPOT to exceed the minimum TPOT SLO of the running requests set $R'$:

$$\text{EstimatedTPOT}(VBS(R'), L_{avg}(R')) \leq \min_{r' \in R(t)} S_{TP}(r')$$

The mechanism guarantees that the admitted requests obey their TPOT SLOs.

# 4 Experiments

In this section, we evaluate our proposed method against the baselines and the effectiveness of each component. We show that our proposed method can achieve state-of-the-art performance in terms of goodput and SLO attainment rate under different scenarios.

## 4.1 Experimental Setup

**Testbed.** We conduct our experiments on a server with 4 NVIDIA A100 GPUs, each with 80GB of memory. The GPUs are interconnected with NVLink between pairs (GPU0-GPU1 and GPU2-GPU3), while communication between pairs utilizes the PCIe fabric and the system's NUMA interconnect.

**Serving Models.** We use Meta Llama-3.1 8B [22] and Google Gemma-2 27b [23] as serving models. All experiments use FP16/BF16 precision, which is the most common setting in LLM deployment. The 8B model runs on a single GPU, and the 27B model runs on 4 GPUs with tensor parallelism [24].

**Workloads.** We evaluate using the ShareGPT [25] and LMSYS-Chat-1M [26] datasets, which are the widely used datasets collected from real-world conversations. For each dataset and model pair, we use the same prompt set to train the predictor for consistency. Regarding the SLO setting, to fully explore the serving heterogeneous requests, we consider requests with 6 categories, as summarized in Table 1. Category 1 represents requests with both tight TTFT and TPOT constraints (e.g., code generation [11]). Categories 2 and 3 represent requests with tight TPOT and relatively loose TTFT constraints (e.g., tool call response). Categories 4 and 5 represent requests with loose TPOT and tight TTFT constraints (e.g., reading-speed responses of chatbot [11]). Category 6 represents requests with both loose TPOT and TTFT constraints (e.g., Summarization [12]). Note that for the 27B model, we loosen the SLO constraints to account for the increased model size.

Table 1: SLO categories for different model sizes.

| Model | Metric | Category | | | | | |
|---|---|---|---|---|---|---|---|
| | | 1 | 2 | 3 | 4 | 5 | 6 |
| Llama-3.1 8B | TTFT (s) | 0.5 | 2 | 3 | 0.5 | 1 | 7.5 |
| | TPOT (ms) | 30 | 30 | 30 | 50 | 50 | 50 |
| Gemma-2 27B | TTFT (s) | 1.0 | 4 | 6 | 1.0 | 2 | 15 |
| | TPOT (ms) | 60 | 60 | 60 | 100 | 100 | 100 |

**Baselines.** We compare our method with the following baselines:

- **vLLM** [4]: A state-of-the-art LLM serving system that uses a throughput-oriented scheduling strategy. The vLLM scheduler prioritizes prefills.

- **S3** [18]: An LLM inference system that predicts the output sequence length and employs the shortest job first scheduling. Since S3 is implemented on basic LLM inference without techniques like PagedAttention, we re-implement its core scheduling strategy within vLLM for a fair comparison.

- **Mooncake** [27]: A production-grade LLM serving platform that employs an early-rejection-based admission control for SLO guarantees. We integrate this mechanism into vLLM.

## 4.2 QPS-Scaling SLO Attainment

We compare SCORPIO's goodput and SLO attainment rate against baselines on ShareGPT and LMSYS-Chat-1M with heterogeneous SLOs, under varying request arrival rates [11, 12]. As shown in Figure 4, SCORPIO achieves higher goodput and SLO adherence rate compared with the baselines, especially at high QPS. For example, at a QPS of 15, it yields up to 8.8-14.4× higher goodput and 40.7-46.5% higher SLO adherence than baselines, demonstrating robust burst handling through its SLO-oriented scheduling. In contrast, vLLM's greedy admission leads to severe TPOT violations. S3's output-length ranking and Mooncake's strict rejection both exhibit suboptimal SLO adherence. Note that at lower QPS, baselines occasionally show better performance than SCORPIO. For example, when serving Gemma2-27b on ShareGPT at a QPS of 5, vLLM achieves a 1.08× goodput and a 1.5% higher SLO adherence rate. One contributing factor is resource contention between the sequence length predictor and the LLM server when co-located on the same GPUs. This can be mitigated by deploying the predictor on a separate low-cost GPU, as studied in Appendix A.5. Alternatively, even without an extra GPU, this could be addressed by detecting low-load conditions and dynamically switching to a simpler, lower-overhead scheduling strategy, which is left as future work.

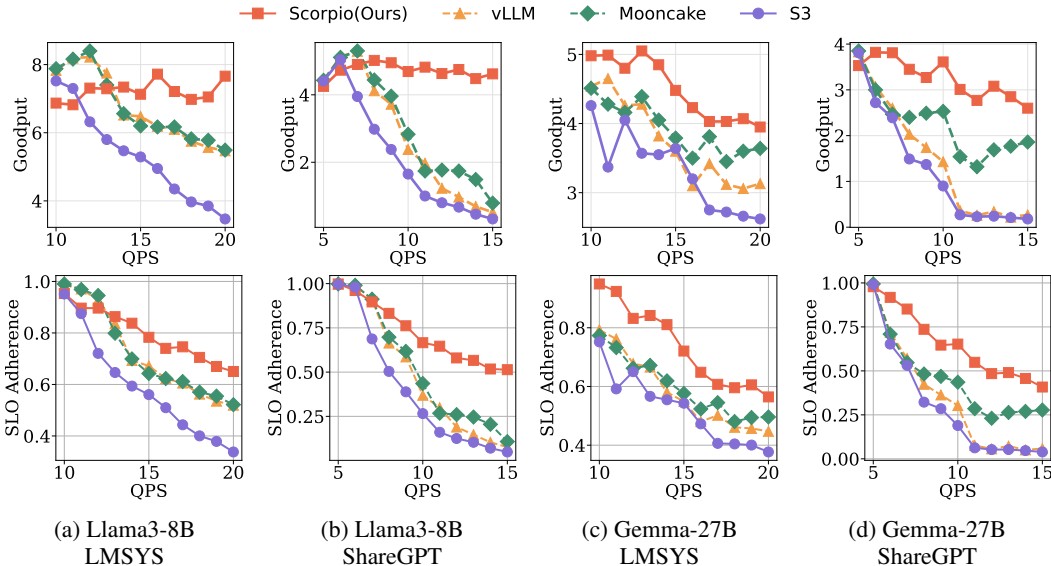

Figure 4: Impact of different scheduling strategies on goodput and SLO adherence vs QPS.

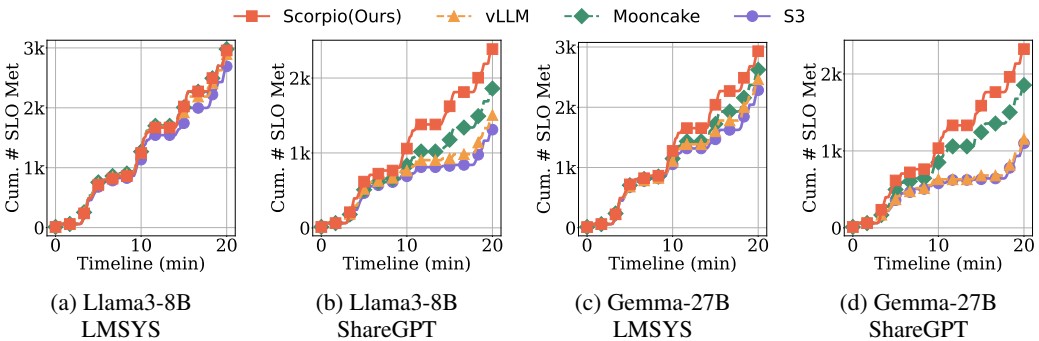

Figure 5: Cumulative number of SLO-met requests over time on the LMSYS and ShareGPT dataset.

## 4.3 Real-World Trace Serving

We further evaluate SCORPIO using 20-minute real-world traces [28], which exhibit periods of both bursty and light load (§A.2). Figure 5 reports the cumulative number of SLO-met requests for Llama3-8B and Gemma2-27B models across both datasets. Across all settings, SCORPIO achieves the highest cumulative SLO-met counts. For example, SCORPIOachieves a 1.25, 2.01, and 2.11× higher SLO-met requests compared with Mooncake, vLLM, and S3, respectively. Throughput-oriented baselines like vLLM exhibit noticeable slowdowns during traffic spikes (e.g., minutes 3-5 and 9-11), caused by uniform batching and greedy admission, leading to widespread TPOT violations. Mooncake (Figure 5b,d) alleviates some pressure via early rejections, but its SLO gains taper off with a too strict admission control mechanism and lack of consideration for the heterogeneous SLOs. S3's length ranking also exhibits suboptimal SLO attainment. In contrast, SCORPIO steadily maintains a lead in cumulative SLO-met requests. By combining TTFT Guard and TPOT Guard, SCORPIO effectively handles the heterogeneous SLOs.

## 4.4 Effectiveness Analysis

**Ablation Study.** For this evaluation, we evaluate the core components of SCORPIO. As shown in Figure 6, we add the proposed components incrementally and assess the performance. For simplicity, we only present results at a QPS of 14. We find that 1) without any component, SCORPIO's goodput is significantly reduced. 2) Including only the TTFT Guard effectively reduces TTFT

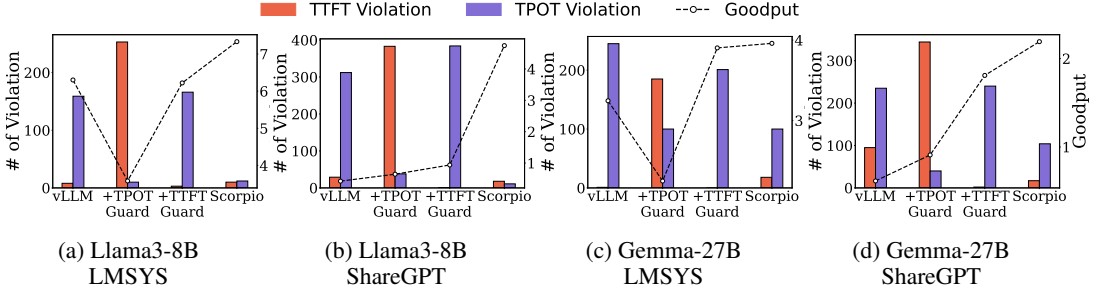

Figure 6: Ablation study on the impact of different scheduling strategies.

Table 2: Overhead of SCORPIO's scheduling

| Model | Dataset | Overall Time (s) | Schedule Time (s) | Scorpio (s) | Overhead (%) |
|-------|---------|------------------|-------------------|-------------|--------------|
| Llama-3-8B | ShareGPT | 58.1 | 1.83 | 0.073 | 0.12% |
| Llama-3-8B | LMSYS-Chat-1M | 58.4 | 2.04 | 0.102 | 0.17% |
| Gemma-27B | ShareGPT | 81.2 | 2.26 | 0.126 | 0.16% |
| Gemma-27B | LMSYS-Chat-1M | 84.9 | 2.08 | 0.097 | 0.11% |

violations as intended, but results in severe TPOT violations, and vice versa. These results show the interdependence of the components, highlighting their importance for the overall performance.

**Overhead of SCORPIO's Scheduling.** We illustrate the overhead of SCORPIO's scheduling in responding to 512 requests, shown in Table 2. The SCORPIO's overhead is measured by summing the time taken by all scheduling sub-components, including the TTFT Guard and TPOT Guard. We find that the overhead is negligible, less than 1% in all settings. We do not account the time of predictor since it is shown to be negligible in previous works [18, 29].

## 5 Related Work

In recent years, LLM serving systems have been widely studied. Orca [6] and vLLM [4] introduce continuous batching and paged attention for efficient GPU VRAM utilization, which have been adopted as ad-hoc strategies for LLM serving [30]. Building on this, subsequent research further optimizes GPU utilization. Sarathi-Serve [7] proposes a piggyback strategy to schedule prefill and decode together. SplitWise [31] and DistServe [12] propose to split the prefill and decode into different instances, preventing interference between batch-like prefill jobs and latency-critical decode tasks. Beyond raw performance, [32] and [33] propose fairness-aware scheduling mechanisms to ensure equitable service among requests. Prediction of request characteristics (e.g., output length) is another significant direction, enabling scheduling policies such as SSJF [18, 34, 19, 29] and SRTF [35]. Recently, adhering to SLOs has gained more focus. Mooncake [27] proposes an early-rejection strategy to handle the overload scenario. QM [10] proposes a queue management framework to improve TTFT SLO adherence. Concurrent work [11, 16] adjusts the allocation of tokens on the fly with speculative decoding [36] to achieve customized SLO serving. In this work, we explore scenarios with heterogeneous SLOs. We propose a fine-grained scheduling strategy that exploits the heterogeneity of SLOs to maximize goodput and SLO adherence.

## 6 Conclusion

In this paper, we present SCORPIO, a novel SLO-oriented LLM serving system designed to maximize system goodput and SLO adherence for requests with heterogeneous SLOs. By exploiting SLO heterogeneity, we employ specialized mechanisms working in concert: a TPOT Guard and a TTFT Guard, supported by an accurate predictor. Evaluations demonstrate that SCORPIO significantly outperforms state-of-the-art throughput-oriented systems, improving system goodput by up to 14.4× and SLO adherence rate by up to 46.5% across various scenarios.

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

## A Appendix

### A.1 Implementation Details

Our codebase contains the complete codebase for SCORPIO, including all experimental harnesses and visualization scripts described in this paper. It is built upon a fork of the vLLM project, which we extended with our scheduling system. We will release the codebase soon.

For the sequence length predictor, we augment the original OPT architecture with a simple linear projection layer that maps the hidden state of the final token to the predicted output length bin. We collect 20K samples from ShareGPT and LMSYS-Chat-1M datasets. We split the dataset into the training set, validation set, and test set with a ratio of 6:2:2. We train the model on the training set using a batch size of 64 for 8 epochs. Following [29], we truncate input prompts to a maximum of 2048 tokens to accommodate OPT's context window constraints. The training data example is shown in Listing 1. Recognizing that traditional accuracy metrics can be misleading with imbalanced class distributions, we introduce multiple evaluation metrics, including off-by-n accuracy (predictions within $n$ bin of the ground truth), Kendall's Tau correlation coefficient and root mean squared error of length prediction for comprehensive assessment (§A.4). Note that we deploy the sequence length predictor in the same GPU with the LLM inference server for resource efficiency. We also study the impact of the interference of the sequence length predictor on the serving system (§A.5). For the analytic models, we profile the step time of the inference of the same training set that is used for the sequence length predictor, ensuring consistency.

Listing 1: Training data example.

```
{
  "prompt": "Translate the following English text to French: 'Hello,
     world!'",
  "output_length": 128,
  "label": 1
}
```

Table 3: Hyperparameters when training the sequence output length predictor

| Hyperparameter | Value |
|---|---|
| Optimizer | Adam |
| Learning Rate | 2e-5 (constant) |
| $\beta_1$ | 0.9 |
| $\beta_2$ | 0.999 |
| Batch Size | 64 |
| Epochs | 8 |

### A.2 Dataset Distribution and Trace Pattern

For Meta Llama3.1-8b, we directly use the dataset provided by [29]. For Gemma2-27b, we randomly sample 20k samples and collect the prompt-response pairs. We show the dataset distribution as in Figure 7. For the input length, we compute the input length by appending the chat template onto the prompts. On average, the Gemma-27B has a slightly shorter input length of about 150 tokens and an output length of around 130 tokens. For the trace pattern used in our Real-World Trace Serving experiment 4.3, we select the first 20 minutes of the Azure Inference Trace [28]. As shown in Figure 8, the trace exhibits periods of both bursty and light load, with the QPS reaching a peak of over 60 requests per second at around 14 minutes from the start of the trace.

### A.3 Analytic Model Accuracy

We report the statistics metrics of the analytic model (including TTFT estimator and TPOT estimator §3.1) for different metrics in Table 4. The results demonstrate excellent model accuracy across all metrics. For example, when using Meta Llama3.1-8b and ShareGPT dataset, the $R^2$ values approaching 1.0 (0.994 for Prefill Time Estimater and 0.987 for Inter-Token Time Estimater) indicate

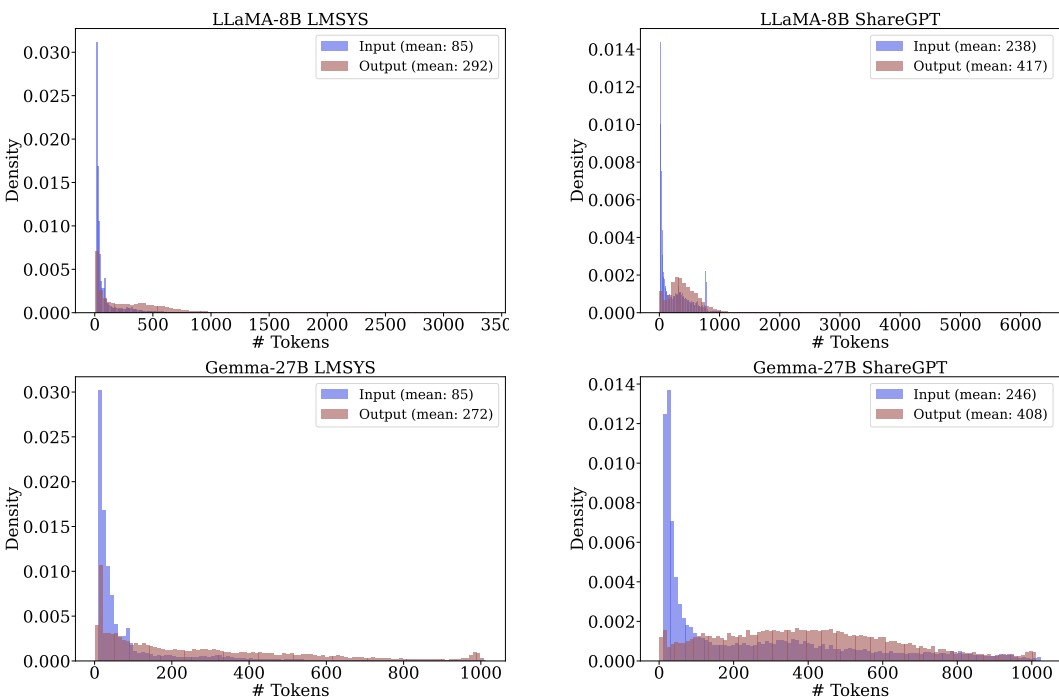

Figure 7: Dataset distribution of different models on ShareGPT and LMSYS.

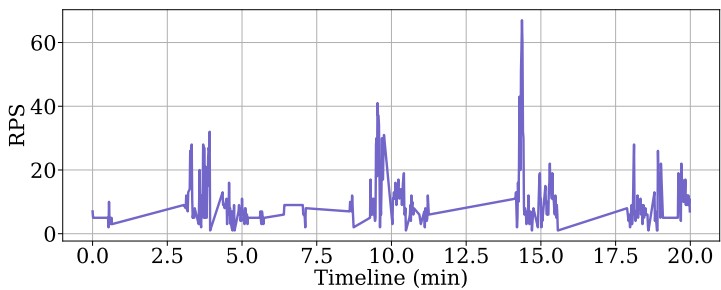

Figure 8: Requests per second over time of first 20 minutes of Azure Inference Trace [28].

that our analytic models explain over 98% of the variance in the data, showing strong predictive power. The low RMSE values (2.07ms for Prefill Time Estimator and 0.871ms for Inter-Token Time Estimater) suggest minimal prediction errors in absolute terms. Additionally, the MAPE values (4.812% for Prefill Time Estimator and 2.3% for Inter-Token Time Estimater) indicate that our predictions are within 5% of the actual values on average, demonstrating high relative accuracy. These results are consistent across both the ShareGPT and LMSYS datasets, showing the robustness of our analytic models.

Table 4: Accuracy of the analytic model for different metrics

| Model | Dataset | Analytic model | $R^2$ ($\uparrow$) | RMSE (ms) ($\downarrow$) | MAPE (%) ($\downarrow$) |
|---|---|---|---|---|---|
| Llama3.1-8B | ShareGPT | Prefill Latency | 0.994 | 2.070 | 4.812 |
| | | Inter-Token Latency | 0.987 | 0.871 | 2.300 |
| | LMSYS | Prefill Latency | 0.983 | 1.281 | 4.664 |
| | | Inter-Token Latency | 0.979 | 0.643 | 2.224 |
| Gemma2-27b | ShareGPT | Prefill Latency | 0.988 | 3.158 | 5.922 |
| | | Inter-Token Latency | 0.953 | 3.534 | 7.859 |
| | LMSYS | Prefill Latency | 0.957 | 3.436 | 8.311 |
| | | Inter-Token Latency | 0.905 | 3.041 | 7.444 |

## A.4 Complete Results of Bucketing Strategy Analysis

In this part, we comprehensively evaluate the bucketing strategies for the sequence length predictor. As shown in Table 5, we designed bucket numbers ranging from 10 to 1000 based on two strategies: 1) equal-width and 2) equal-frequency. In the former, we evenly divide the sequence length range into equal-width buckets, while in the latter, we evenly divide the sequence length based on the output length distribution of the datasets. Different from previous works [18, 19] that only report the classification accuracy, we also evaluate the off-by-n accuracy (predictions within $n$ bin of the ground truth), Kendall's tau coefficient to measure the relative order accuracy and root mean square error (RMSE).

Our analysis revealed several key insights. For equal-width bucketing, we find that using a moderate number of bins (e.g., 100) provides the best balance between multiple performance metrics. For example, when serving Llama3-8B on ShareGPT, using a number of 100 bins achieves 0.54 in Kendall's tau and 195.8 in RMSE. Also, note that even if the exact accuracy is low, the off-n accuracy remains relatively high. This suggests the model effectively places predictions close to the true bucket. In contrast, using too few buckets, which are commonly used in previous works, shows very low tau and high RMSE despite misleading high accuracy. This discrepancy stems from the highly skewed data distribution, where over 93% of samples concentrate in the first bucket. Too many buckets also degrade the prediction performance. For equal-frequency bucketing, all binning configurations show high RMSE, indicating unsuitability for absolute length prediction. This is because with imbalanced data, equal-frequency bucketing leads to initial bins being narrow and later bins excessively wide, making precise classification difficult. Based on the analysis, we adopt equal-width bucketing and set the number of buckets to 100 for sequence length predictor in all experiments (§4).

Table 5: Performance comparison of Equal-width and Equal-frequency bucketing with different bucket numbers on ShareGPT and LMSYS datasets using Llama3-8B and Gemma-27B.

| Model | Strategy | # Buckets | ShareGPT | | | | | LMSYS | | | | |
|---|---|---|---|---|---|---|---|---|---|---|---|---|
| | | | Tau (↑) | Acc. (%) | Off-1 Acc. (%) | Off-2 Acc. (%) | RMSE (↓) | Tau (↑) | Acc. (%) | Off-1 Acc. (%) | Off-2 Acc. (%) | RMSE (↓) |
| Llama3-8B | Equal-width | 10 | 0.25 | 88.7 | 99.9 | 100.0 | 301.4 | 0.25 | 95.8 | 100.0 | 100.0 | 304.9 |
| | | 20 | 0.45 | 70.0 | 97.6 | 99.6 | 247.1 | 0.50 | 79.4 | 99.0 | 99.9 | 222.3 |
| | | 50 | 0.54 | 43.2 | 84.0 | 95.0 | 202.4 | 0.60 | 56.3 | 85.4 | 94.0 | 201.5 |
| | | 100 | **0.54** | 27.0 | 61.3 | 79.7 | **195.8** | 0.61 | 41.2 | 67.1 | 81.0 | 196.2 |
| | | 200 | 0.51 | 15.4 | 37.2 | 54.5 | 199.4 | **0.62** | 29.7 | 49.9 | 61.6 | 197.5 |
| | | 500 | 0.51 | 7.6 | 17.8 | 27.3 | 201.3 | 0.62 | 18.9 | 33.7 | 42.0 | **193.2** |
| | | 1000 | 0.49 | 5.0 | 11.2 | 16.4 | 207.9 | 0.61 | 13.6 | 24.0 | 30.4 | 196.7 |
| | Equal-frequency | 10 | 0.54 | 28.1 | 59.8 | 78.1 | 1088.7 | 0.65 | 38.1 | 72.4 | 86.8 | 1171.4 |
| | | 20 | 0.53 | 16.2 | 37.7 | 53.8 | 1042.7 | 0.64 | 25.7 | 50.6 | 66.8 | 837.9 |
| | | 50 | 0.52 | 8.4 | 19.2 | 28.6 | 689.9 | 0.63 | 14.5 | 28.6 | 39.6 | 361.3 |
| | | 100 | 0.51 | 5.0 | 11.4 | 17.0 | 652.4 | 0.62 | 10.4 | 17.4 | 24.1 | 287.5 |
| | | 200 | 0.48 | 2.7 | 5.8 | 8.9 | 474.4 | 0.60 | 8.8 | 12.2 | 16.1 | 418.1 |
| | | 500 | 0.44 | 2.0 | 3.2 | 4.3 | 994.5 | 0.57 | 7.7 | 8.7 | 10.1 | 263.3 |
| | | 1000 | 0.42 | 1.6 | 1.9 | 3.0 | 277.0 | 0.52 | 7.6 | 7.8 | 8.1 | 237.2 |
| Gemma-27B | Equal-width | 10 | 0.21 | 94.0 | 100.0 | 100.0 | 249.5 | 0.22 | 96.9 | 100.0 | 100.0 | 288.8 |
| | | 20 | 0.52 | 74.9 | 99.6 | 100.0 | 195.3 | 0.47 | 79.4 | 98.7 | 100.0 | 221.9 |
| | | 50 | **0.62** | 52.0 | 88.8 | 97.8 | 157.2 | 0.63 | 57.6 | 86.3 | 95.9 | 177.7 |
| | | 100 | 0.61 | 34.8 | 68.2 | 83.9 | **156.5** | **0.65** | 43.6 | 69.6 | 81.8 | 176.6 |
| | | 200 | 0.60 | 21.2 | 48.5 | 63.4 | 154.2 | 0.64 | 31.4 | 53.0 | 65.0 | **169.6** |
| | | 500 | 0.59 | 11.5 | 26.6 | 38.4 | 156.9 | 0.64 | 18.3 | 34.2 | 44.3 | 170.9 |
| | | 1000 | 0.57 | 6.8 | 15.8 | 23.0 | 161.6 | 0.63 | 14.5 | 25.0 | 32.6 | 171.4 |
| | Equal-frequency | 10 | 0.62 | 36.7 | 68.6 | 83.6 | 1253.8 | 0.67 | 38.2 | 71.0 | 86.2 | 949.2 |
| | | 20 | 0.61 | 22.3 | 48.2 | 63.0 | 914.6 | 0.66 | 26.6 | 51.8 | 66.3 | 865.2 |
| | | 50 | 0.60 | 11.5 | 25.8 | 37.8 | 489.9 | 0.64 | 14.5 | 29.4 | 39.8 | 503.2 |
| | | 100 | 0.59 | 5.7 | 14.5 | 21.9 | 476.9 | 0.64 | 9.8 | 18.6 | 25.4 | 552.9 |
| | | 200 | 0.56 | 3.2 | 8.0 | 12.6 | 250.4 | 0.61 | 7.4 | 11.3 | 15.9 | 235.0 |
| | | 500 | 0.52 | 2.0 | 4.2 | 5.8 | 343.0 | 0.57 | 6.6 | 8.1 | 9.7 | 268.2 |
| | | 1000 | 0.51 | 1.5 | 1.8 | 2.8 | 192.7 | 0.56 | 6.2 | 6.6 | 7.2 | 221.2 |

## A.5 Interference of the Sequence Length Predictor on the Scheduling System

Since simply deploying the sequence length predictor and LLM server on the same GPUs may lead to resource contention, we study the interference of the sequence length predictor on the serving system in this part. As shown in Figure 9, we compare the goodput and slo adherence of the system

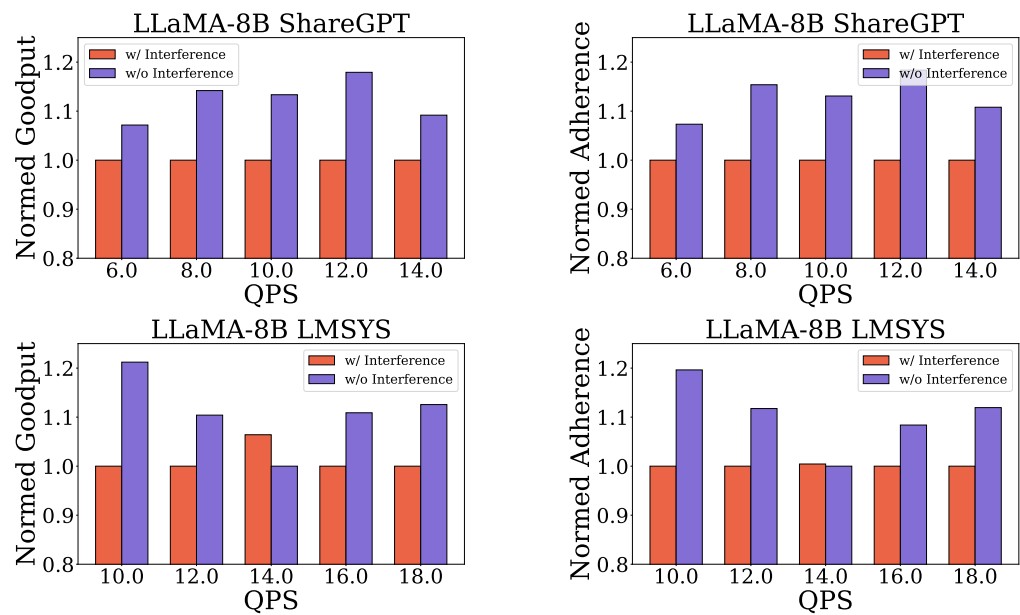

Figure 9: Interference of the sequence length predictor on the serving system.

with and without the interference of the sequence length predictor. For w/ interference, we deploy the predictor and the LLM server on the same A800 GPUs. For w/o interference, we deploy the predictor on a separate GTX 3090 GPU with 24GB memory. On average, we find that the interference of the sequence length predictor leads to a 5% - 20% performance degradation on goodput and slo adherence on average.

## A.6 Limitations

The proposed scheduler currently works with standard LLM serving techniques such as continuous batching and paged attention. How to integrate the scheduler with the latest optimizations, such as prefill-decode disaggregation [12], is an interesting direction for future work. Also, as we mentioned in the §4.2, the scheduler shows a bit of performance degradation at low QPS. This is because while SCORPIO's complexity is advantageous under heavy load, the intervention of such control degrades performance at low QPS. This calls for a more flexible and adaptive scheduling strategy that could switch to a simpler, lower-overhead scheduling method. We leave these as future work.

