# OpenReview forum: "Scorpio: Serving the Right Requests at the Right Time for Heterogeneous SLOs in LLM Inference"
_NeurIPS.cc/2025/Conference — Submitted to NeurIPS 2025_

### Official Review · Reviewer_EdBo · 2025-06-22

**Clarity:** 3
**Significance:** 3
**Originality:** 2
**Rating:** 4
**Confidence:** 4

**Summary:**

This paper proposes a heterogeneous SLO-oriented serving system, SCORPIO, that actively exploits per-request latency targets (TTFT and TPOT) rather than treating every prompt identically. The system couples lightweight sequence-length prediction with two cooperating control loops—TTFT Guard (least-deadline-first re-ordering plus early rejection) and TPOT Guard (virtual-batch-size admission plus credit-based batching). Across workloads, SCORPIO boosts goodput versus existing baselines, while keeping the scheduling overhead low.

**Questions:**

Please refer to the previous comments.

**Ethical Concerns:**

["NO or VERY MINOR ethics concerns only"]

**Final Justification:**

Authors address my concerns in their rebuttal regarding predictor setup and generality. However, I am still not impressed by the novelty of this paper, as to it core, the length predictor is studied by many other works. The modification to that is just some binning strategy. The overall contribution of this paper is incremental to me, so I decided to keep the weak accept score for this.

**Limitations:**

Yes

**Quality:**

3

**Strengths And Weaknesses:**

Strengths:
1. This paper focuses on a timely problem, moves beyond throughput to heterogeneous latency guarantees, a gap in mainstream systems like vLLM.
2. The paper is well written and easy to follow
3. This paper proposes a system with modular design, two orthogonal guards plus a tiny predictor; each module’s contribution is dissected via ablation.
4. Evaluation is thorough with multiple models (8 B/27 B), two public traces, and QPS scaling experiments substantiate the claims. It also evaluates against three representative schedulers covering greediness, length-based ranking, and admission-control paradigms.

Weaknesses:
1. This paper follows a similar design decision, which is to introduce an output sequence length predictor. However, it’s not clear that the fine-tuned OPT-125 M would be able to generalize across different datasets. Do you need to train a separate predictor for each dataset?
2. As shown in this paper, previous works are focusing on similar problems, and the core of this line of work is the length predictor. Including this paper, they are all sharing the same methodology (length prediction) with papers only considering homogeneous SLOs, which makes it hard to judge relative novelty. The remained components are quite trivial, which is just the earliest deadline first. Can you give more justification for why such a length predictor would even work? What’s new and special about your length predictor? Especially, when you compare with S3, is the performance gap mainly due to the different performance of the length predictor? What’s S3’s performance if it uses your length predictor?
3. In the main evaluation, there are cases where Scorpio yields lower goodput at relatively lower request rates. Can you elaborate on that?

---

> ### Author Rebuttal · Authors · 2025-07-30
>
> We thank you for the insightful and helpful feedback\! We would like to address your questions in the response below.
>
> **Q1: On the generalization of the length predictor across different datasets and whether a separate predictor is needed for each.**
>
> **A1:** To clarify, in our current experiments, we train a separate predictor for each dataset, which is a fast process (approx. 10 minutes). To further investigate its cross-dataset generalization capability (as suggested by the Reviewer), we conducted a new experiment by training a predictor exclusively on one dataset and evaluating it on a different dataset. Note that these two datasets have different distributions (see Figure 7 in our paper).
>
> The results are summarized in Table 3 shown below. We report Kendall's Tau (↑) , Off-2 Accuracy  (↑) and RMSE  (↓),  where  arrows indicate the direction for better performance. Definitions of the metrics are in Appendix A.4 in our paper.
>
> **Table 3:  generalization study of the length predictor**
>
> | Method              | │ |  |     Tested on ShareGPT         |         | │ |  |      Tested on LMSYS        |         |
> | ------------------- |:-:| :----------------: | :----------: | :-----: |:-:| :-------------: | :----------: | :-----: |
> |                     | │ | Tau ↑             | Off-2 Acc ↑ | RMSE ↓ | │ | Tau ↑          | Off-2 Acc ↑ | RMSE ↓ |
> | Trained on ShareGPT | │ | 0.56               | 0.80         | 195.8   | │ | 0.37            | 0.67         | 257.3   |
> | Trained on LMSYS    | │ | 0.46               | 0.48         | 244.9   | │ | 0.65            | 0.81         | 196.1   |
>
> As shown in Table 3, the predictor experiences a performance degradation when evaluated on an out-of-distribution dataset, which can be attributed to the significant distribution shift between the two datasets (as shown in Figure 7 in our paper). For example, when trained on ShareGPT and tested on LMSYS, Kendall's Tau drops from 0.56 to 0.37, and Off-2 Accuracy decreases from 0.80 to 0.67. However, the predictor still retains significant predictive capability, demonstrating a certain level of generalization. For instance, the Kendall's Tau remains well above random, indicating it can still effectively rank requests by length.
>
> Furthermore, in a real-world production environment, this distribution shift can be effectively managed by periodically retraining the predictor on recent historical data, a standard practice to maintain model performance \[1, 2\]. We will add a detailed discussion of this new experiment to the Appendix.
>
> &nbsp;
>
> **Q2.1: On the overall novelty of our work beyond length prediction, given that prior work uses similar methods for homogeneous SLOs.**
>
> **A2.1:** We would like to argue that the novelty of our paper goes well beyond a length-prediction work with minor enhancements. Our main contribution lies in a novel scheduling system designed for a new and challenging problem: **serving LLM requests with heterogeneous, multi-metric SLOs (both TTFT and TPOT).** To the best of our knowledge, no prior work has holistically addressed this scenario. While the sequence predictor plays an important, supportive role, the core contribution lies in our system-level co-design of the **TTFT Guard** and, most significantly, the **TPOT Guard**.
>
> * For the TPOT Guard, we introduce two key mechanisms. First, inspired by CPU scheduling, we designed a credit-based batching system that fairly allocates processing opportunities according to each request's specific TPOT SLO. Furthermore, inspired by load control in cloud computing, we developed a VBS-based admission control mechanism. This is crucial for preventing the cascading SLO violations common in prior systems under high load.
> * For the TTFT Guard, we use a Least-Deadline-First (LDF) strategy, which is simple but highly effective for this SLO dimension. Additionally, we implement the Unattainable SLO Rejection strategy to proactively reject requests that will inevitably miss their deadlines under continuous high load scenarios. Also, this could be efficient for mitigating the starvation problem common in scheduling system (See our response to R2, Q2).
>
> Both guards are guided by dedicated analytic models to make informed, real-time decisions. We believe this integrated approach represents a substantial and novel contribution to the field.
>
> **Q2.2: On the specific contributions and novelty of our length predictor.**
>
> **A2.2:** Regarding the sequence predictor itself, while we share a similar architecture with prior work like S3, our approach is far from a naive application. We observed that the binning strategies used in previous work were suboptimal for real-world data distributions. To address this, we conducted a comprehensive experimental study (detailed in Appendix A.4 in our paper) to determine the most effective binning strategy, a new contribution that improves prior approaches.
>
> **Q2.3: On the source of performance gain over the S3 baseline and whether it's mainly due to the predictor.**
>
> **A2.3:** To clarify, for a fair comparison, **our S3 baseline was implemented using our improved predictor.** Hence, the performance improvement in our results (Figure 4 in our paper) stems from our overall scheduling system (TTFT/TPOT Guards) over S3's simple Shortest-Job-First strategy for the complex, heterogeneous SLO problem. This is further validated by our ablation study (Figure 6 in our paper), which clearly shows each component of SCORPIO makes a substantial contribution to the final performance.
>
> &nbsp;
>
> **Q3: On why SCORPIO sometimes yields lower goodput at lower request rates.**
>
> **A3:** This is an important trade-off that we also observed and discussed in Section 4.2. A key contributing factor to this performance characteristic is the resource contention between our sequence length predictor and the main LLM server when they are co-located. Our study in Appendix A.5 confirms this, showing that this performance degradation can be mitigated by deploying the predictor on a separate, low-cost GPU.
>
> &nbsp;
>
> **Reference:**
>
> \[1\] Zhang, Haoran, et al. "Why did the model fail?" attributing model performance changes to distribution shifts. ICML. 2023\.
>
> \[2\] Chen C, Li R, Hu Y, et al. Overcoming Catastrophic Forgetting by Exemplar Selection in Task-oriented Dialogue. ACL. 2024.

---

> > ### Comment · Reviewer_EdBo · 2025-08-03
> > **Response**
> >
> > Thanks authors for your rebuttal. Most of my conceptual questions have been resolved.

---

> ### Author Response · Authors · 2025-08-04
> **Official Comment by Authors**
>
> Dear Reviewer EdBo,
>
> Thank you for your response. We are glad that we could address most of your concerns and appreciate your thoughtful review. We will incorporate the key discussions and results into our final revised manuscript.
>
> Best Regards,
>
> Paper 13264 Authors

---

### Official Review · Reviewer_s6zD · 2025-06-28

**Clarity:** 2
**Significance:** 3
**Originality:** 2
**Rating:** 4
**Confidence:** 3

**Summary:**

The paper introduces SCORPIO, an LLM serving system designed to handle heterogeneous Service Level Objectives (SLOs), specifically addressing TTFT and TPOT. The authors identify that current LLM serving systems prioritize throughput at the expense of meeting diverse latency requirements across different requests. To tackle this, SCORPIO leverages adaptive scheduling using two specialized mechanisms:

- **TTFT Guard:** Implements least-deadline-first reordering and rejects requests unlikely to meet TTFT.
- **TPOT Guard:** Uses Virtual Batch Size (VBS)-based admission control and a credit-based batching mechanism tailored to TPOT constraints.

Additionally, SCORPIO employs a sequence length predictive module to support decision-making. Empirical evaluations show that SCORPIO significantly improves system goodput (up to 14.4×) and SLO adherence (up to 46.5%) compared to state-of-the-art systems.

**Questions:**

Please see Weaknesses.

**Ethical Concerns:**

["NO or VERY MINOR ethics concerns only"]

**Final Justification:**

This is a qualified paper, and the author has answered my questions well. I have decided to maintain my score.

**Limitations:**

yes

**Quality:**

3

**Strengths And Weaknesses:**

# Strengths:
- **Quality:** Rigorous evaluation demonstrating significant improvements over existing methods, with well-chosen baseline comparisons.
- **Clarity:** Clearly structured and detailed explanations of methods and system components, supported by informative diagrams and algorithms.
- **Significance:** Directly addresses a highly relevant problem in LLM inference serving—heterogeneous SLO management—which is crucial in practical deployment scenarios.

# Weaknesses
- **SLO setting concerns:** SCORPIO is a system specifically designed to enhance SLO attainment and goodput. Different SLOs should have a significant impact on experimental results, but there is no appropriate basis for setting SLOs during the experiment (although the authors have set many SLOs and conducted ablation experiments). I suggest that the authors search for authoritative SLO setting guidelines to set the SLOs for the experiment.
- **Starvation concerns:** This paper does not explicitly discuss or handle the starvation problem. The scheduling method described in the paper might cause a potential starvation problem: The TTFT Guard will place requests with closer deadlines at the front. Under continuous high-load scenarios, requests with looser TTFT requirements may be repeatedly delayed, failing to receive timely service, thus causing starvation.TPOP Guard is designed to use credit accumulation to determine the frequency of request batching, in extreme cases, when strict TPOT SLO requests keep pouring in, the rate at which loose TPOT requests accumulate credit will significantly slow down, which may lead to requests failing to obtain service opportunities for a long time, thus causing a situation of request starvation.
Therefore, I think that the SCORPIO scheduling system, although it effectively improves the satisfaction rate of heterogeneous SLOs, indeed has potential starvation risks under high-load and long-term operation scenarios.The author does not explicitly discuss this issue in the paper and does not propose any related mitigation strategies. Subsequent research may need to further explore how to better balance the relationship between throughput, SLO achievement rate, and request fairness to reduce the risk of starvation.

---

> ### Author Rebuttal · Authors · 2025-07-30
>
> We thank you for the insightful and helpful feedback\! We would like to address your questions in the response below.
>
> **Q1: On the basis of setting experimental SLOs, suggesting a search for more authoritative guidelines.**
>
> **A1:** We apologize that our rationale for the SLO settings was not sufficiently clear. Our SLOs were, in fact, carefully chosen to be representative of realistic use cases, guided by prior work. Specifically, the TPOT requirements for interactive tasks (Categories 1, 4, 5\) were inspired by the performance benchmarks in \[1\] (reference \[11\] in our paper). Regarding TTFT, while paper \[1\] does not explicitly define its values for these cases, we set TTFTs to reflect typical user expectations (e.g., a tight TTFT for an interactive chatbot). Similarly, our summarization task (Category 6\) was informed by \[2\] (reference \[12\] in our paper). The remaining categories (2 and 3\) were intentionally designed to introduce diversity in SLO profiles. We will revise the manuscript to explicitly provide this detailed justification, clarifying how our settings were grounded in and inspired by existing literature to ensure the soundness of our experimental design.
>
> &nbsp;
>
> **Q2: On the potential for request starvation under high load and the lack of explicit discussion or mitigation strategies.**
>
> **A2:** We agree that request starvation is a critical aspect, and more discussion is needed. We will clarify this in our revision by discussing the potential for starvation in both the TTFT and TPOT guards and how our proposed mechanisms address them.
>
> * **Regarding the TTFT Guard:** Our Least-Deadline-First (LDF) reordering inherently prevents starvation. A request with a looser TTFT SLO is not perpetually delayed; as time passes and its deadline approaches, its priority dynamically increases, eventually moving it to the front of the queue to be served before its deadline expires. In cases of continuous high load where even this dynamic prioritization is insufficient, our Unattainable SLO Rejection strategy (Section 3.3, second paragraph) acts as a final backstop, ensuring a bounded wait by proactively rejecting requests that will inevitably miss their deadlines rather than allowing them to starve. Furthermore, as discussed in the paper (Section 3.3, second paragraph), these rejected requests can be handled with alternative approaches common in cloud computing, such as migration to other nodes or elastic scaling \[3, 4, 5\].
> * **Regarding the TPOT Guard:** Starvation is prevented by a two-level mechanism. Our VBS-based Admission Control carefully manages the volume of concurrent requests, working in concert with the Credit-based Batching mechanism. This synergy ensures that all admitted requests are served harmoniously, each adhering to its specific TPOT SLO. While some requests are inevitably rejected under high load, this is a deliberate design choice that aligns with our core focus: *maximizing the serving quality and goodput for admitted requests, rather than attempting to serve all requests under limited resources*.
>
> In summary, SCORPIO is designed with mechanisms that explicitly prevent starvation through dynamic prioritization and proactive rejection. We believe that further research exploring the nuanced balance between goodput, SLO attainment, and request fairness is a promising and practical direction, which we will leave for our future work.
>
> &nbsp;
>
> **References**
>
> \[1\] Li Z, Chen Z, Delacourt R, et al. Adaserve: Slo-customized llm serving with fine-grained speculative decoding. arXiv. 2025.
>
> \[2\] Zhong Y, Liu S, Chen J, et al. {DistServe}: Disaggregating prefill and decoding for goodput-optimized large language model servin, OSDI, 2024\.
>
> \[3\] Suresh Chandra Moharana, Bishwabara Panda, Manoj Kumar Mishra, et al. Load balancing in virtualized environments using virtual machine migration: A comprehensive survey. International Journal of Knowledge-based and Intelligent Engineering Systems, 2021\.
>
> \[4\] Chen Y, Peng Y, Bao Y, et al. Elastic parameter server load distribution in deep learning clusters. *ACM Symposium on Cloud Computing,* 2020\.
>
> \[5\] Gu D, Zhao Y, Zhong Y, et al. Elasticflow: An elastic serverless training platform for distributed deep learning, ASPLOS, 2023\.

---

> > ### Comment · Reviewer_s6zD · 2025-08-04
> >
> > My concerns are mostly addressed in the rebuttal. Thanks to the authors.

---

> > > ### Author Response · Authors · 2025-08-04
> > > **Official Comments by Author**
> > >
> > > Dear Reviewer s6zD,
> > >
> > > Thank you for your feedback and for acknowledging our rebuttal. We appreciate your valuable comments throughout the review process and will ensure that the clarifications provided are reflected in our revised manuscript.
> > >
> > > Best Regards,
> > >
> > > Paper 13264 Authors

---

### Official Review · Reviewer_xrDr · 2025-07-01

**Clarity:** 3
**Significance:** 3
**Originality:** 3
**Rating:** 4
**Confidence:** 2

**Summary:**

The paper introduces SCORPIO, an SLO-oriented serving system for Large Language Models (LLMs) designed to maximize system throughput and adherence to Service Level Objectives (SLOs) for requests with heterogeneous SLOs. It incorporates two specialized mechanisms, TPOT Guard and TTFT Guard, supported by a predictive module to ensure high system goodput and optimized SLO adherence. SCORPIO improves system performance by up to 14.4× and enhances SLO adherence by 46.5%, demonstrating its superior performance over traditional throughput-oriented approaches.

**Questions:**

Please refer to the weakness section.

**Ethical Concerns:**

["NO or VERY MINOR ethics concerns only"]

**Final Justification:**

The response has addressed the majority of my concern, therefore, I remain my scoring as acceptance. However, it is woth noting that the appendix code seems to contain author's name, which is a violation of the double blind policy.

**Limitations:**

Yes

**Quality:**

3

**Strengths And Weaknesses:**

### Strengths

- The paper addresses a practical and meaningful problem: SLO heterogeneity for adaptive scheduling in LLM serving.
- The paper is clearly written, with effective figure illustrations that help convey key ideas.
- Extensive experimental results and a thorough overhead analysis are provided, demonstrating the effectiveness of the proposed method.

### Weaknesses
- Adaptability to "long output" or "long input" scenarios: While the paper effectively addresses SLO heterogeneity, it would be valuable to discuss the applicability of the proposed framework to two iconic LLM serving scenarios: "reasoning" (which involves long output tokens) and "long input" (which involves long input tokens). Assessing how well the framework handles these scenarios is crucial for its broader practical usage.

Despite the paper provides valid contribution, the supplementary materials seem to violate the double-blind rule. The author's code contained at least 21 files with the author's name, “tingfeng,” appearing 143 times. Based on reference [20], One could infer that one of the authors is Tingfeng Lan.

---

> ### Author Rebuttal · Authors · 2025-07-30
>
> We thank the reviewer for this insightful suggestion.
>
> **Q1: On the framework's applicability to iconic LLM serving scenarios with long inputs or long outputs (e.g., reasoning).**
>
> **A1**: We agree that assessing our framework in "long output" (e.g., reasoning) and "long input" (e.g., document analysis) scenarios is crucial for evaluating its practical utility. To address this, we have conducted two new sets of experiments.
>
> - For the long output scenario, we evaluated our method on the opencoder-reasoning benchmark \[1\]. We curated a subset of requests with lengthy generated responses (1K-2K tokens), which are significantly longer than datasets used in our main evaluation (see Figure 7 in our paper).
> - For the long input scenario, we used the dataset published in \[2\] to simulate long-input tasks, selecting prompts with lengths between 1K and 2K tokens.
>
> The results for SLO adherence are presented below in Table 1 and Table 2. They show that our solution, SCORPIO, consistently outperforms all baselines in both scenarios, maintaining significantly higher SLO adherence, especially as the system load (QPS) increases. For example, at a QPS of 5, Scorpio yields up to 43%-50% higher SLO adherence than baselines in the long output scenario, and 34%-38% higher in the long input scenario. This confirms that our scheduling mechanisms are robust and effective even for these challenging scenarios, which is consistent with the findings in our paper.
>
> **Table 1: SLO Adherence vs. QPS (Long Output Scenario)**
>
> | Method                   | │ |            |                |                |    QPS             |                |                |                |
> | ------------------------ |:-:| -------------- | :------------- | :------------- | :------------- | :------------- | :------------- | :------------- |
> |                          | │ | **2.0**        | **2.5**        | **3.0**        | **3.5**        | **4.0**        | **4.5**        | **5.0**        |
> | Mooncake                 | │ | 0.99           | **0.97**       | 0.88           | 0.69           | 0.55           | 0.27           | 0.17           |
> | vLLM                     | │ | 0.99           | 0.96           | 0.88           | 0.69           | 0.54           | 0.29           | 0.10           |
> | S3                       | │ | 0.99           | 0.97           | 0.88           | 0.68           | 0.53           | 0.26           | 0.10           |
> | **Scorpio (Ours)**      | │ | **0.99**       | 0.96           | **0.88**       | **0.81**       | **0.71**       | **0.66**       | **0.60**       |
>
> &nbsp;
>
> **Table 2: SLO Adherence vs. QPS (Long Input Scenario):**
>
> | Method                   | │ |           |                |                |         QPS         |                |                |                |
> | ------------------------ |:-:| -------------- | :------------- | :------------- | :------------- | :------------- | :------------- | :------------- |
> |                          | │ | **2.0**        | **2.5**        | **3.0**        | **3.5**        | **4.0**        | **4.5**        | **5.0**        |
> | Mooncake                 | │ | 0.91           | 0.46           | 0.23           | 0.18           | 0.13           | 0.12           | 0.11           |
> | vLLM                     | │ | 0.83           | 0.56           | 0.26           | 0.16           | 0.13           | 0.10           | 0.07           |
> | S3                       | │ | **0.94**       | 0.45           | 0.24           | 0.14           | 0.10           | 0.08           | 0.07           |
> | **Scorpio (Ours)**      | │ | 0.90           | **0.79**       | **0.70**       | **0.65**       | **0.58**       | **0.54**       | **0.45**       |
>
> We will add the full results (e.g., goodput) and a detailed analysis to the appendix of the final manuscript.
>
> &nbsp;
>
> **Q2: On a potential anonymity issue in the supplementary materials.**
>
> **A2:** We assure the Area Chair that we hold the integrity of the double-blind review process in the highest regard. This was a genuine technical oversight. The name appeared in auto-generated dependency files (.dep) created by the vLLM framework's build system, which unfortunately embeds the absolute file paths of the build environment. We are deeply sorry for this mistake and assure the AC and reviewers that this was an honest error and not an attempt to circumvent the rules.
>
> &nbsp;
>
> **References**
>
> \[1\] Ahmad W U, Narenthiran S, Majumdar S, et al. Opencodereasoning: Advancing data distillation for competitive coding, arXiv preprint. 2025\.
>
> \[2\] Cohan A, Dernoncourt F, Kim D S, et al. A discourse-aware attention model for abstractive summarization of long documents. NAACL. 2018\.

---

> > ### Comment · Reviewer_xrDr · 2025-08-05
> > **Response to author response**
> >
> > Thanks the authors for the feedback, the response has addressed the majority of my concern.

---

> ### Author Response · Authors · 2025-08-05
> **Official Comments by Author**
>
> Dear Reviewer xrDr,
>
> Thank you for your feedback. We appreciate your valuable suggestions throughout the review process and will ensure all discussed improvements are incorporated in our final manuscript.
>
> Best Regards,
>
> Paper 13264 Authors

---

### Decision · Program_Chairs · 2025-09-17

**Decision:**

Reject

**Comment:**

The paper proposes SCORPIO, an SLO-aware LLM serving system that integrates TTFT Guard, TPOT Guard, and a predictive module to exploit heterogeneous latency requirements for improved scheduling. The system shows substantial improvements in goodput and SLO adherence compared with existing methods.

The strengths of the paper include a timely problem formulation, clear system design, extensive experiments, and well-presented analysis.

However, reviewers consistently noted weaknesses: the reliance on length predictors that echo prior work limits novelty, SLO settings lack authoritative grounding, starvation risks under high load are insufficiently addressed, and some observed goodput drops raise concerns. One reviewer also highlighted a double-blind violation in the supplementary code, which is a procedural issue. While the rebuttal addressed technical concerns by adding new experiments on long input/output scenarios and clarifying starvation and predictor generalization, the responses did not fundamentally resolve the incremental novelty issue, and the anonymity breach remains problematic.

In weighing the contributions, I find the work technically competent but not sufficiently novel or robust for NeurIPS acceptance. Therefore, I recommend rejection.